# Mechanical properties of a biodegradable self-expandable polydioxanone monofilament stent: *In vitro* force relaxation and its clinical relevance

Ales Bezrouk[1]*, Tomas Hosszu[2], Ludek Hromadko[3,4], Zuzana Olmrova Zmrhalova[3], Martin Kopecek[1], Martin Smutny[1], Iva Selke Krulichova[1], Jan M. Macak[3,4], Jan Kremlacek[1]

1 Department of Medical Biophysics, Faculty of Medicine in Hradec Kralove, Charles University, Hradec Kralove, Czech Republic, 2 Department of Neurosurgery, University Hospital Hradec Kralove, Hradec Kralove, Czech Republic, 3 Center of Materials and Nanotechnologies, Faculty of Chemical Technology, University of Pardubice, Pardubice, Czech Republic, 4 Central European Institute of Technology, Brno University of Technology, Brno, Czech Republic

* bezrouka@lfhk.cuni.cz

**Data Availability Statement:** All relevant data are within the manuscript and its (S1–S5 Files).

## Abstract

Biodegradable stents are promising treatments for many diseases, *e.g.*, coronary artery disease, urethral diseases, tracheal diseases, and esophageal strictures. The mechanical properties of biodegradable stent materials play a key role in the safety and efficacy of treatment. In particular, insufficient creep resistance of the stent material could result in premature stent collapse or narrowing. Commercially available biodegradable self-expandable SX-ELLA stents made of polydioxanone monofilament were tested. A new, simple, and affordable method to measure the shear modulus of tiny viscoelastic wires is presented. The important mechanical parameters of the polydioxanone filament were obtained: the median Young's modulus was $\tilde{E}$ = 958 (922, 974) MPa and the shear modulus was $\tilde{G}$ = 357 (185, 387) MPa, resulting in a Poisson's ratio of v = 0.34. The SX-ELLA stents exhibited significant force relaxation due to the stress relaxation of the polydioxanone monofilament, approximately 19% and 36% 10 min and 48 h after stent application, respectively. However, these results were expected, and the manufacturer and implanting clinician should be aware of the known behavior of these biodegradable materials. If possible, a biodegradable stent should be designed considering therapeutic force rather than initial force. Additionally, new and more advanced biodegradable shape-memory polymers should be considered for future study and use.

## Introduction

Stenting has long been a conventional medical intervention. This technique is typically used to reopen occluded hollow tubular body structures. Stenting is usually minimally invasive, resulting in its increasing use. Implantation of a coronary stent is one of the most common medical

**Funding:** We acknowledge the financial support from the Ministry of Education, Youth and Sports of the Czech Republic: o programme UK PROGRES Q40-09 o project LQ1601 o project CZ.02.1.01/0.0/0.0/17_048/0007421 o project LM2018103 Authors who received salaries from the funder: programme UK PROGRES Q40-09: Ales Bezrouk, Martin Kopecek, Iva Selke Krulichova project LQ1601: Jan Macak project CZ.02.1.01/0.0/0.0/17_048/0007421: Ales Bezrouk, Ludek Hromadko, Jan Macak project LM2018103: Jan Macak, Ludek Hromadko, Zuzana Olmrova Zmrhalova The funders had no role in study design, data collection and analysis, decision to publish, or preparation of the manuscript.

**Competing interests:** The authors have declared that no competing interests exist.

interventions to reopen an occluded vessel [1]. First-generation stents were metallic. Although metallic stents are effective in preventing acute occlusion and reducing late restenosis, many concerns remain [2]. The most notable limitations are a chronic local inflammatory reaction due to permanent implantation of a foreign body, restriction of vascular vasomotion due to a metal cage, and the risk of late and very late stent thrombosis [3]. Additionally, for the treatment of malignant or benign esophageal strictures, metallic stents have shown controversial results [4–6]. These complications can be severe and may require additional treatment. Restenting can be very complicated or even impossible [7]. These safety and efficacy limitations have triggered much research into developing biodegradable stents and more potent drug delivery systems [1].

Biodegradable stents have been successfully used in the treatment of coronary artery diseases [1–3,8] and esophageal strictures [6,9–11]. Biodegradable stents may also be utilized for urethral, tracheal, and other applications [12]. When compared with first-generation metallic stents, the main benefits of biodegradable stents are the absence of the long-term complications of temporary scaffolding, since they are able to dissolve in the patient's body [10,12,13] and have improved mechanical and chemical biocompatibility [12]. However, these benefits are accompanied by certain disadvantages resulting from the specific mechanical properties of biodegradable materials, especially with respect to the mechanical stability required for a certain treatment duration. Therefore, the mechanical properties of biodegradable stent materials play a key role in treatment safety and efficacy. These factors are important for proper stent design, application, and treatment.

Many studies have investigated stent material properties. These studies have mainly focused on biocompatibility [14], corrosion resistance [12,13,15–17], "simple elastic" stress-strain characteristics [13,18,19], and elastic memory [16,20–22]. Other studies have investigated or developed new materials [23–25] and stent designs [13,22,25].

However, a very important material property related to the stent's mechanical behavior is viscoelasticity. This parameter primarily affects the mechanical stability of the stent material over time, and therefore, its ability to withstand external forces for a long duration, as described by creep resistance [23] or stress relaxation [26]. Creep resistance and stress relaxation are directly related to the collapse pressure [25,27]. Creep resistance must be maintained because a stent is subjected to radial pressure from the supported tissue, and insufficient creep resistance could result in premature stent collapse or narrowing [23]. The stent material viscoelasticity has a large influence on the stent parameters and should be taken into account when choosing an appropriate stent.

One of the very few commercially available biodegradable stents is the self-expandable esophageal SX-ELLA stent made of polydioxanone monofilament. We hypothesize that SX-ELLA stents will exhibit significant force relaxation from polydioxanone monofilament stress relaxation due to viscosity. To the best of our knowledge, no studies have investigated the viscoelastic properties of polydioxanone monofilament. The goal of this study is to test the force relaxation of a polydioxanone monofilament stent and to test other important properties, such as Young's modulus and shear modulus, that are crucial for proper and safe stent design and application.

## Materials and methods

For the *in vitro* tests, we used commercially available biodegradable self-expandable SX-ELLA stents (Fig 1) made of polydioxanone monofilament. These are 32 branch braided stents with atraumatic ends. Each stent (Fig 1) is 80 mm long ($L_N$), 25 mm in diameter ($D_1$), with short 31 mm diameter funnel-shaped ends ($D_2$), and made of a single monofilament.

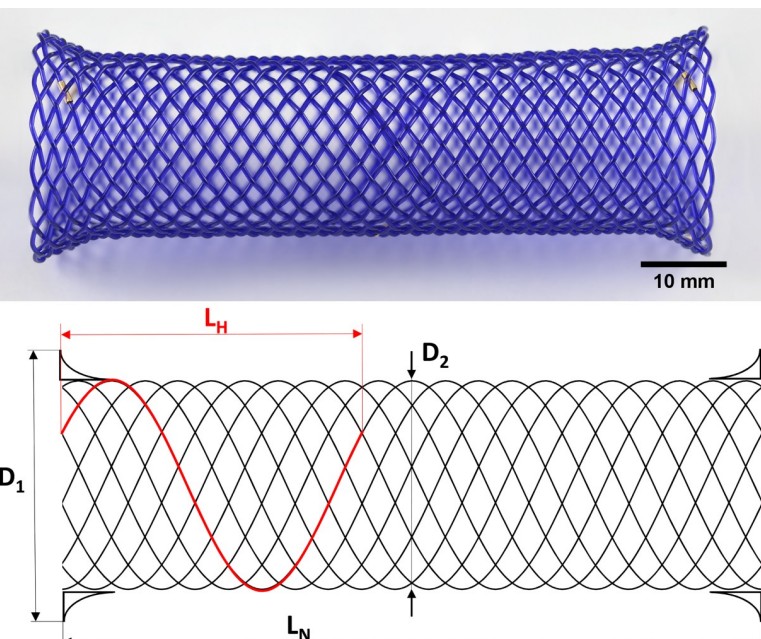

**Fig 1.** A) photograph and B) schematic drawing of the biodegradable self-expandable SX-ELLA stents. $L_N$ is the total nominal length of the unloaded stent; $D_1$, $D_2$ are the nominal stent diameters; $L_H$ is the length of a single turn of a single helix of the unloaded stent (helix length).

We used a new stent for each test. We used 10 stents for the *in vitro* stent force relaxation test, 20 sample wires for estimating the Young's modulus, 25 sample wires for estimating the shear modulus, 2 samples for thermal gravimetric analysis (TGA) including 1 sample for thermal gravimetric analysis and gas chromatography–mass spectrometry (TGA-GCMS), and 2 samples for scanning electron microscopy (SEM) analysis. For the Young's and shear moduli estimations, TGA, TGA-GCMS, and SEM, we acquired all the samples (sample wires) from new stents by removing the funnel-shaped ends and disassembling.

## Material evaluation

First, we checked the structure and chemical composition of the stent material. We used TGA and TGA-GCMS to determine the chemical composition and SEM analysis to assess the material structure.

During TGA, we heated the samples to 600˚C at a rate of 10˚C/min. The samples were measured in an open platinum crucible in a synthetic air atmosphere (20 ml/min). The samples were measured in an air atmosphere to determine polydioxanone's thermal stability in this atmosphere. TGA was performed on a Pyris 1 TGA (PerkinElmer, USA).

During TGA-GCMS, we heated the sample to 300˚C at a rate of 10˚C/min. The samples were measured in an open platinum crucible in a nitrogen atmosphere (20 ml/min; Linde N2 4.6). In TGA-GCMS, a nitrogen atmosphere must be used to protect the mass spectrometer's heated wolfram filament from oxygen. The TGA-GCMS was performed on a Pyris 1 TGA, GC Clarus 680, and MS Clarus SQ 8T (all PerkinElmer, USA).

We inspected the filament material's homogeneity, looking at the internal structure using SEM investigation. The samples were prepared by mechanically rupturing the filaments by simply cracking them with tweezers, mounting them on the SEM stub using carbon adhesive tape, and directly (without an additional coating to prevent charging) visualizing them using

field-emission SEM (FE-SEM JEOL JSM 7500F, Japan) and the in-lens secondary electron detector.

## Stent geometry

Knowledge of stent geometry parameters is important for design, clinical practice, and further mechanical analyses. From a clinical perspective, the most important information is the dependence of stent radial pressure $p_R$ on stent deformation. The radial force $F_R$ is then used as an important parameter for stent production output quality control. However, it is difficult to accurately measure these radial parameters. Typical significant problems with direct radial measurements include the mechanical interaction between the measuring device and the stent and the effect of friction. These negative effects cannot be sufficiently determined and thus effectively and accurately compensated for. Instead, the easier and more accurate measurements of stent axial force and deformation can be used [28]. The effects of friction and device-stent interactions in the axial measurements are negligible. The measured data can then be mathematically transformed into clinically relevant radial pressure and deformation data. We can also simplify the stent geometry by neglecting the short funnel-shaped ends. Then, using the approach and modifying the equations of Zahora *et al.* [28], we can derive the relationship between stent radial pressure $p_R$ and the stent's measured experimental parameters–axial force $F_A$ and instantaneous total length $L_S$:

$$p_R = 4\pi \frac{L_N}{L_H} \frac{F_A}{L_S^2} \tag{1}$$

Similarly, we can derive the relationship between radial force and axial force and length:

$$F_R = 4\pi F_A \sqrt{\frac{\left(\frac{\pi^2 D_1^2}{L_H^2} + 1\right) L_N^2}{L_S^2} - 1} \tag{2}$$

In Eqs (1) and (2), $L_H$ is the length of a single turn of a single helix of the unloaded stent (helix length). Furthermore, in Eqs (1) and (2), the total length of the stent $L_S$ is a measured parameter that changes with the applied load, while the nominal length of the stent $L_N$ and the helix length $L_H$ are the nominal parameters of an unloaded stent, which remain constant during measurement. A detailed derivation of Eqs (1) and (2) is presented in S1 File. We performed 20 measurements to determine the helix length. To determine the Young's and shear moduli of the stent material, we performed 20 measurements of the stent wire diameter $d$. We used the Extol 3426 caliper (Madal Bal a. s., Czech Republic) for these measurements.

## Young's modulus

To further determine the stent material properties, we measured the Young's moduli of 20 stent wire samples using ordinary tensile tests performed on an Instron 3343 with a 1 kN force transducer and Instron's special pneumatic grips for wire measurements. We performed the measurements in a temperature-controlled chamber at a constant temperature of $(37 \pm 0.3)°$C. We also monitored the relative humidity for extreme fluctuations, *i.e.*, a difference of more than 10%, due to polydioxanone's natural hydrophilicity that could potentially affect the measured data. We used a standard measurement procedure, such as those reported by Greenwald *et al.* [29] and Kreszinger *et al.* [30]. We gathered the force-extension data, which we then further transformed to standard stress-strain curves using the sample wire diameter $d$ and the original effective length of the sample $L_{TS}$. We performed 20 measurements of $L_{TS}$ using the

Extol 3426 caliper (Madal Bal a. s., Czech Republic). We determined the Young's modulus as the slope of the greatest pseudolinear part of the stress-strain curve.

## Shear modulus

The shear modulus is a very important stent design parameter. However, we did not find any values for the shear modulus $G$ of polydioxanone in current published literature. Since the shear modulus measurement of tiny viscoelastic wires is nontrivial, and current conventional measurement methods and devices cannot be used, we developed our own dedicated tool, shown in Fig 2, with a modified measurement method.

The measured specimen (Fig 2D) is fixed between two Jacobs microchucks; one of the chucks (Fig 2B) is fixed to a solid frame, and the other chuck (Fig 2B) revolves and is connected to a pair of pulleys. During the measurement, the wire connected to the Instron force transducer (Fig 2E) is pulled, as shown by the arrow in Fig 2, revolving the Jacobs chuck (Fig 2C) and twisting the measured specimen. The shift and force applied to the pulling wire are measured. We performed the measurements in a temperature-controlled chamber at a constant temperature of $(37 \pm 0.3)°C$. We also monitored the relative humidity for extreme fluctuations, *i.e.*, a difference of more than 10%, due to polydioxanone's natural hydrophilicity that could potentially affect the measured data.

The measurement of a thin wire with substantial viscous properties is nontrivial. The viscous properties cause plastic deformation of the wire, preventing it from returning to its original shape when released. The plastic deformation, in connection with the inner friction of the measurement tool, considerably affects the measured data. We solved these issues by using an elastic reverting element, denoted by the thick gray dashed line in Fig 2. This elastic element allows the system specimen-tool-Instron to revert back to its original position when the pulling force is released. This component also measures the "background force profile", *i.e.*, measuring the force dependence on the elongation when using the tool without a specimen, thus eliminating the influence of the tool's inner friction by subtracting the background force profile from the measured data (measurement with a specimen). An overview with detailed photos and videos of the dedicated tool with the measured specimen are provided in S2 File.

First, we calculated the median of the slope of the pulling part of the background force profile $\tilde{K}_{FB}$ (25 measurements taken for the tool **without a specimen**; maximum shift of 45 mm). Next, we determined the background force $F_{B90}$ (Eq (3)) at a shift corresponding to a 90° rotation in the revolving chuck-pulling pulley part of the tool.

$$F_{B90} = \tilde{K}_{FB} \frac{\pi \bar{D}_{RP}}{4} \tag{3}$$

where $-D_{RP}$ is the mean diameter of the "pulling" pulley obtained from 20 measurements using the Extol 3426 calipers (Madal Bal a. s., Czech Republic).

In the same way, we calculated the median of the slope of the pulling part of the total force profile $\tilde{K}_{FT}$ (measurement of 25 specimens; maximum shift of 45 mm) and the respective total force $F_{T90}$ at a shift corresponding to a 90° rotation in the revolving chuck-pulling pulley part of the tool. Due to the high impact of the background on our measurements, we statistically validated the difference between the slopes of the background and the total force profile.

Then, we calculated the torque $M$ needed to twist (only) the wire specimen by 90° (Eq (4))

$$M = \bar{D}_{RP} \frac{F_{T90} - F_{B90}}{2} \tag{4}$$

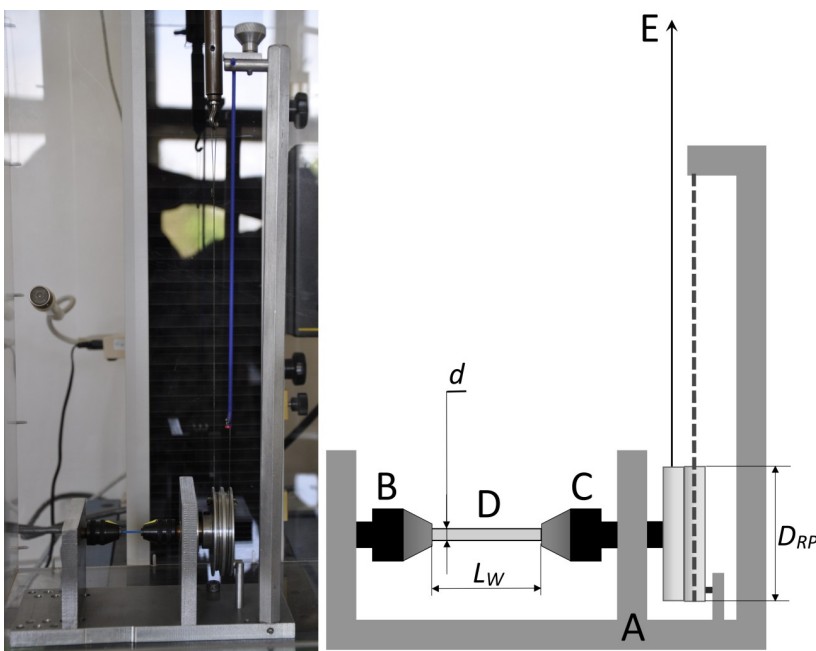

**Fig 2. Photograph and scheme of the dedicated tool for shear modulus measurement.** The device consists of a solid frame denoted by A, a fixed Jacobs microchuck denoted by B, a revolving Jacobs microchuck, denoted by C, tightly connected to a pair of pulleys, and a pulling wire, denoted by E, connected to the "pulling" pulley on one side and the Instron force transducer on the other side (marked by the arrow). The tested specimen (wire) is denoted by D. The thick gray dashed line denotes the elastic reverting element connected to the "reverting" pulley on one side and to the solid frame on the other side. $D_{RP}$ is the diameter of the "pulling" pulley; $d$ is the stent wire diameter; $L_W$ is the specimen (wire) length.

and the second moment of the area of the circular cross-section $J_T$ (Eq (5))

$$J_T = \frac{\pi \bar{d}^4}{32}$$

(5)

where $-d$ is the mean stent wire diameter.

Finally, the shear modulus $G$ of the specimen material can be calculated as follows:

$$G = \frac{M \cdot \bar{L}_W}{\frac{\pi}{2} J_T}$$

(6)

where $-L_W$ is the mean specimen length (20 measurements using the Extol 3426 caliper (Madal Bal a. s., Czech Republic)) and $\frac{\pi}{2}$ represents the 90˚ twisting angle.

### *In vitro* stent force relaxation test

The purpose of the *in vitro* force relaxation tests was to separately study the effect of the visco-elasticity of the stent material on the radial force and pressure with respect to esophageal anatomy. The normal esophagus diameter in the human population is approximately 20–33 mm [31]. A symptomatic benign stenosis is clinically variable but is often under 13 mm and rarely over 20 mm in diameter [31]. Therefore, we chose 17 mm as an average symptomatic diameter of an esophageal stricture. An axial stent elongation of 56 mm caused a change in our stent diameter from the nominal 25 mm to the desired 17 mm. The relationship between the length and the diameter of a braided stent was published by Zahora *et al.* [28], and it is also shown in

. The force-relaxation tests were performed on an Instron 3343 with a 1 kN force transducer in a temperature-controlled chamber. Each stent was fixed at its ends by a pair of custom holding tools, as shown in Fig 3. Each stent end was inserted into the tool's cylinder cavity and fixed using the stainless-steel needles, which were threaded transversely through the meshes at the stent ends. This allowed the stent to move relatively freely and easily change its diameter without significant friction.

Then, each stent was extended by 56 mm (+70%) from its nominal length ($L_N$ = 80 mm) to a total length of $L_S$ = 136 mm, at 10 mm·s$^{-1}$ and held in the extended position for at least 48 h (172 800 s) at a constant temperature (37 ± 0.3)˚C. The axial force of the stent was recorded and sampled with respect to the expected dynamics of the relaxation process at the following frequencies: 1/0.1 s$^{-1}$ for the first 25.6 s, 1/3 s$^{-1}$ for the next 60 s, 1/10 s$^{-1}$ for the next 450 s, and 1/300 s$^{-1}$ for the next 172 800 s. We tested the statistical significance of the force relaxation between selected consecutive time steps (0 s, 10 min, 20 min, 30 min, 1 h, 2 h, 24 h, and 48 h). We also monitored the relative humidity for extreme fluctuations, *i.e.*, a difference of more than 10%, due to polydioxanone's natural hydrophilicity that could potentially affect the measured data. For verification and prediction, we also modeled the force relaxation using the generalized Maxwell model, described by Eq (7):

$$F_A(t) = F_\infty + \sum_n F_n e^{-\frac{t}{\tau_n}} \tag{7}$$

where $F_A$ represents the relaxation of the stent's axial force over time, $t$ is the relaxation time, $F_\infty$ is the axial force of the stent at infinite time, *i.e.*, the theoretical steady axial force after the end of the relaxation process, $n$ is the number of partial processes contributing to the total axial force relaxation, $F_n$ is the magnitude of the contribution of the partial process to the total relaxation, and $\tau_n$ is the relaxation time constant of the partially contributing process.

## Statistics

The results were compared, processed, and statistically analyzed using MS Excel 2016 (Microsoft Corp., Redmond WA, USA) and NCSS 10 statistical software (2015, NCSS, LLC., Kaysville, Utah, USA, ncss.com/software/ncss). We used the D'Agostino Omnibus Test to test the normality of the data distribution. We described the data from normally distributed populations using the mean and standard deviation of the sample ($-X$ ± SD), while we described other data using the median and the 1$^{st}$ and the 3$^{rd}$ quartiles of $\tilde{X}$ (1$^{st}$Q, 3$^{rd}$Q). Since the normality of the distribution of the force relaxation data was rejected, we opted to use the Wilcoxon signed-rank test because it does not presuppose a normal distribution. To adjust for multiple comparisons and keep the familywise α at 0.05, we used the Bonferroni correction. The resulting α for a single comparison was 0.007.

## Results

### Material evaluation

The scanning electron micrographs in Fig 4 show the filament morphology. The filament surface shown in Fig 4A clearly displays relatively regular roughness in its longitudinal direction, which is caused by its production and processing. The smooth compact areas in the filament fractures in Fig 4B and 4C show homogenous polydioxanone monofilament without any signs of discrete internal structure. The noticeable cracks in the surface of the filament fracture shown in Fig 4B and partially in Fig 4C are typical of homogenous material fractures.

The TGA results for the 1$^{st}$ ($m_0$ = 2.783 mg) and 2$^{nd}$ ($m_0$ = 2.785 mg) samples of the monofilament stent wire in a synthetic air atmosphere show an initial weight loss (1–1.6)% in the

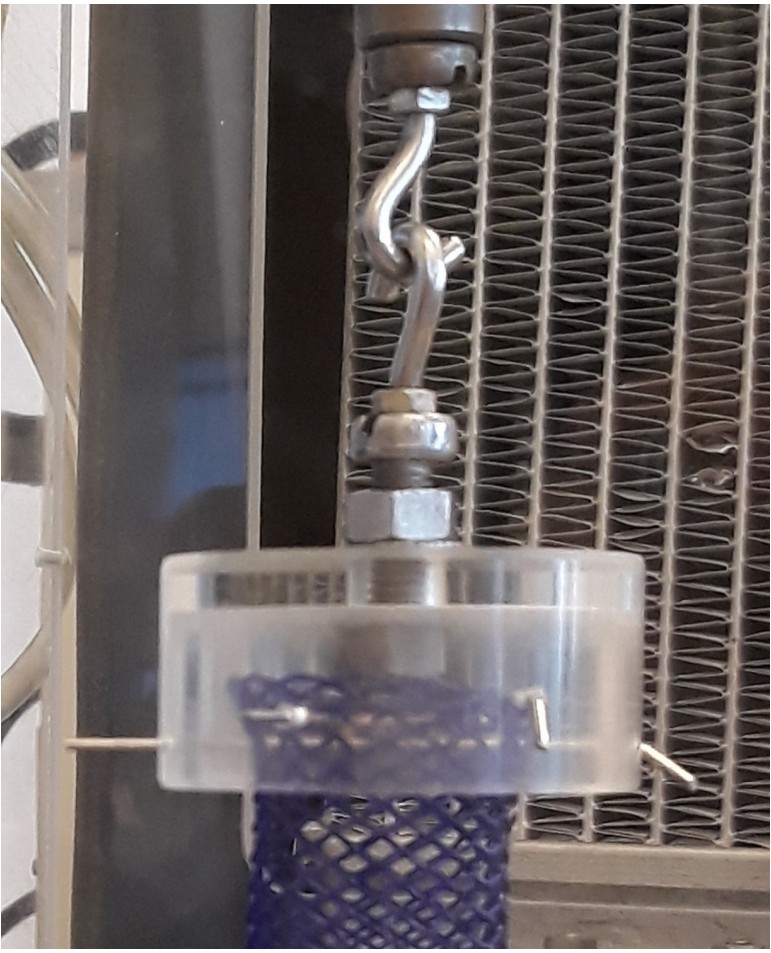

**Fig 3. Details of the custom holding tool.** The tool consists of a hollow cylinder with one open end and one closed end. A hook is mounted to the closed end, allowing its attachment to the measuring device. There are three small holes in the cylinder shell for the transverse attachment of stainless-steel needles.

temperature range of (40–130)°C, most likely due to the loss of internal sample moisture. In the temperature range of (160–300)°C, the weight decreases to approximately (1.2–1.3)% of the original sample weight due to the polymer's thermal decomposition. Derivative thermal gravimetric (DTG) peaks are observed at 245°C and 244°C. TGA of the 3rd sample ($m_0$ = 2.086 mg) in a nitrogen atmosphere, shown in Fig 5, shows that the maximum weight decreases to approximately 3% of the original sample weight in the temperature range of (160–240)°C, with the DTG peak at 234°C. The TGA results correspond to previously published values for polydioxanone by Wang *et al.* [32] and Huang *et al.* [33]. We did not observe any decrease in thermal stability of the polydioxanone filament in a synthetic air atmosphere compared to a nitrogen atmosphere.

During TGA of the 3rd sample, the released gas was taken to a gas chromatograph coupled to a mass spectrometer using a heated transfer line. The chromatogram of the 3rd sample (Fig 6) shows a single peak, which means that only a single substance was detected during the thermal degradation of the sample. Subsequently, the molecular ion peak in the measured mass spectrum shown in the inset of Fig 6 has a value of m/z = 102, which corresponds to the molecular weight of a dioxanone monomer.

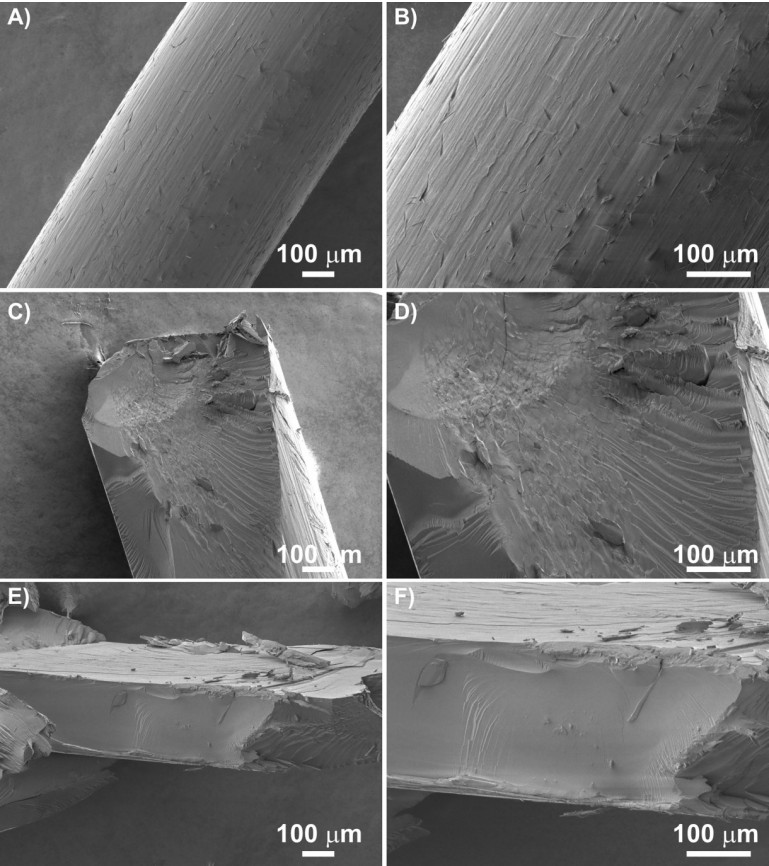

**Fig 4.** Scanning electron micrographs of polydioxanone filament: A) surface at magnifications of 100x and 200x; B) and C) fracture at magnifications of 100x and 200x. The white line corresponds to a size of 100 μm.

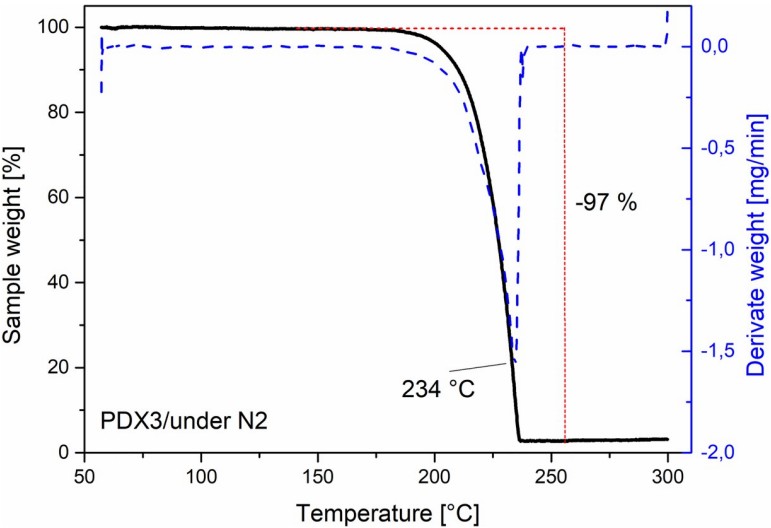

**Fig 5. TGA results for the 3ʳᵈ monofilament stent wire sample.** The black line shows the dependence of the relative sample weight on the temperature. The blue dashed line shows the dependence of the rate of the change of the sample weight (derivate weight) on the temperature. The derivative thermal gravimetric (DTG) peak is observed at 234˚C.

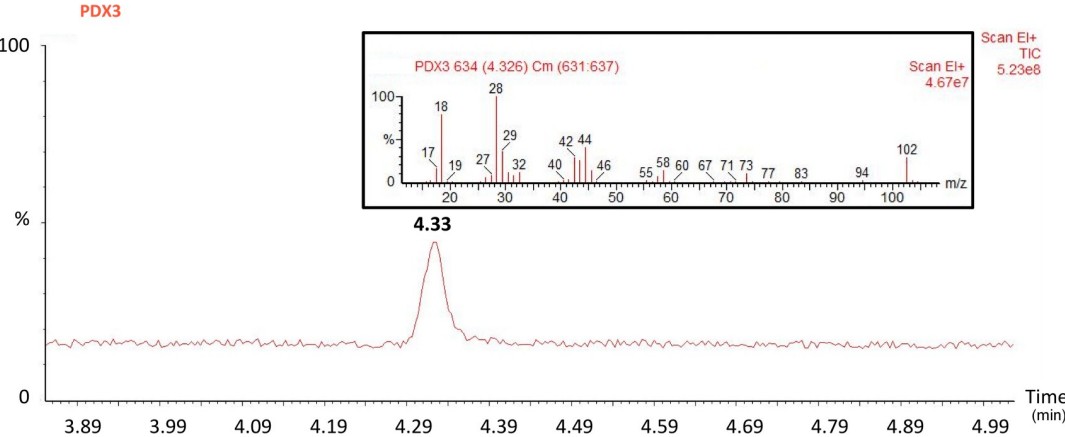

**Fig 6. Chromatogram and mass spectrum (shown in the inset) for the 3$^{rd}$ monofilament stent wire sample from GCMS analysis.** The red curve in the chromatogram shows the relative intensity, with a single peak detected at 4.33 min. The maximum relative intensity (100%) is derived from the maximum peak, which corresponds to the carrier gas $N_2$ (not shown). The red vertical lines in the mass spectrum in the inset show the relative intensity for the detected ions. The maximum relative intensity (100%) is derived from the maximum peak m/z = 28, which corresponds to the carrier gas $N_2$. The molecular ion peak has a value of m/z = 102, which corresponds to the molecular weight of a dioxanone monomer.

## Stent geometry

In addition to the nominal stent dimensions given by the producer ($L_N$, $D_1$, and $D_2$), we found the mean helix length $-L_H = (41.50 \pm 0.61)$ mm and the mean stent wire diameter $-d = (0.64 \pm 0.02)$ mm. The raw data are provided in S3 File.

## Young's modulus

We determined the median Young's modulus of the polydioxanone wire $\tilde{E} = 958$ (922, 974) MPa. We provide the raw data, an example of processed data for Sample 1, and the stress-strain curve for Sample 1 in S4 File.

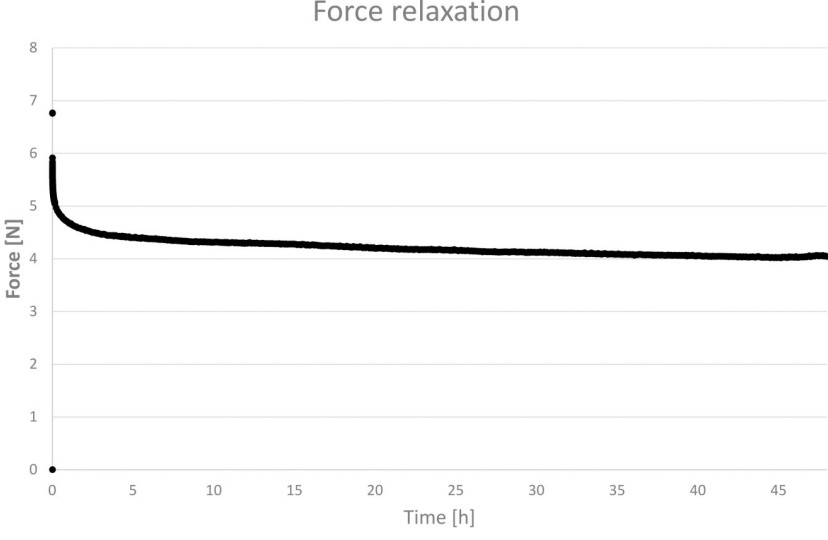

**Fig 7. Force relaxation over time for the selected stent sample.** The samples were measured in an air-conditioned chamber at a constant temperature of $(37 \pm 0.3)°C$. The difference between the maximum and minimum relative humidity values did not exceed 10%.

## Shear modulus

We calculated the median slopes of the pulling part of the background force profile $\tilde{K}_{FB}$ = 18.19 (18.01, 18.28) N·m$^{-1}$ and of the total force profile $\tilde{K}_{FT}$ = 18.83 (18.39, 19.06) N·m$^{-1}$. We found a statistically significant difference ($P < 0.001$) between the slopes of the background and the total force profile, *i.e.*, the increased force values when measured with the sample are not random. We demonstrated that the developed method has sufficient sensitivity.

We found the mean diameter of the "pulling" pulley, $-D_{RP}$ = (52.31 ± 0.03) mm and the mean specimen length, $-L_W$ = (13.20 ± 0.02) mm. Finally, using Eqs (3), (4), (5) and (6), these values result in the shear modulus of the specimen material $\tilde{G}$ = 357 (185, 387) MPa. The raw data is presented in S5 File.

## *In vitro* stent force relaxation test

As shown in Figs 7 and 8, the highest force relaxation rate is observed in the first minutes after load application. The force relaxations are statistically significant at all time steps when compared with the previous time step, as shown in Fig 8. The p-values for the time steps (10 min, 20 min, 30 min, 1 h, 2 h, 24 h, and 48 h) are all $P = 0.003$. The samples were measured at a constant temperature of (37 ± 0.3)°C; the difference in the observed relative humidity never exceeded 10%.

We obtained the best fit to the relaxation data using the generalized Maxwell model (Fig 8) assuming two partially contributing processes. The primary relaxation process was fast, with a relaxation time constant $\tau_1$ = 4.50 min and a relaxation magnitude $F_1$ = 1.19 N. The secondary relaxation process was slower, with a relaxation time constant $\tau_2$ = 189.9 min and a relaxation magnitude $F_2$ = 0.90 N. The axial force at an infinite time limit was $F_{\infty}$ = 3.76 N.

## Discussion

Stenting is an important medical method in which the implanted stent exerts a force on the surrounding tissue to reopen occluded hollow tubular body structures. In clinical practice, considering many factors, such as the patient's anatomy, specific diagnosis, and specific treatment, it is the sole responsibility of the physician to select a force appropriate for the patient. Then, with respect to the desired force, the physician selects a suitable stent with the appropriate parameters. Therefore, the parameters of the stent and its material are crucial.

All the findings from SEM, TGA, and TGA-GCMS agree with the manufacturer's information on the product label, *i.e.*, the stent is made of a single polydioxanone monofilament. The TGA results for the 3$^{rd}$ sample measured in a nitrogen atmosphere corresponds to the data published by Wang *et al.* [32] and Huang *et al.* [33], who used a similar experimental setup and a heating rate of 10°C/min. However, other studies [34,35] show different TGA results, which may be caused by a different experimental setup, heating rate or material type (*e.g.*, lower molecular weight polymer chains) and its possible postproduction processing. For technical reference, certain additional rheological properties of polydioxanone can be found in the study published by Ahlinder *et al.* [34]. Moreover, with regard to the median Young's modulus of the stent wire, $\tilde{E}$ = 958 (922, 974) MPa, the mechanical properties of the wire material are most likely to correspond to those of PDS- or PDX-type polydioxanone [29]. However, it is necessary to consider material processing, which may have significantly changed the material's original properties during stent production.

The shear modulus is very important for proper stent design [28]. For that purpose, we developed a new method to measure the shear modulus of tiny viscoelastic wires. We demonstrated that this method has sufficient sensitivity, despite the relatively small difference

# Force relaxation

**Fig 8. Force relaxation of all samples at selected time steps.** The samples were measured in an air-conditioned chamber at a constant temperature of $(37 \pm 0.3)\,^{\circ}$C. The difference between the maximum and minimum relative humidity values did not exceed 10%. The asterisk (*) indicates the statistically significant force relaxation of the respective time step compared to the previous time step. The red plus sign (+) indicates the calculated force value using the generalized Maxwell model.

between the determined median slopes of the pulling part of the background force profile $\tilde{K}_{FB}$ = 18.19 (18.01, 18.28) N·m$^{-1}$ and the total force profile $\tilde{K}_{FT}$ = 18.83 (18.39, 19.06) N·m$^{-1}$. The determined shear modulus of the polydioxanone material of the biodegradable SX-ELLA stents was $\tilde{G}$ = 357 (185, 387) MPa, which to our knowledge is previously unpublished information. Using the Young's $E$ and shear $G$ moduli, we derived Poisson's ratio, $\nu = 0.34$, which is close to that of certain other polymers, such as PLA (approximately 0.36). This finding also supports the validity of our method.

The force-relaxation test shows statistically significant force relaxations at all time steps compared to the previous step. Notwithstanding the statistical significance results, the clinically relevant information is the force relaxation during the first 10 min (approximately 19%; Fig 8) and the overall force relaxation after 48 h (approximately 36%; Fig 8). Such a loss in force and hence radial pressure supporting the treated tissue could eventually result in premature stent collapse or narrowing [23]. However, it can also be assumed that this loss could be partially compensated [20], probably due to polydioxanone's hydrophilicity. The higher force

data variability at the start of the measurement– 5.85 (5.22, 6.11) N compared with the measurements after 24 h– 3.77 (3.54, 4.19) N and 48 h– 3.75 (3.36, 4.08) N is probably due to the specific production and storage process.

To avoid chemical degradation of the stent material, and considering the storage conditions recommended by the stent manufacturer, we performed all the mechanical measurements in an environment with a low relative humidity (approx. 30%). Furthermore, to eliminate the effect of temperature, we performed all the mechanical measurements at a constant "body" temperature of $(37 \pm 0.3)$˚C. The TGA in an air atmosphere showed that the polydioxanone filament is stable at 37˚C and does not thermally degrade. This allowed us to separately study the effect of stent material viscoelasticity on the radial force and pressure. Moreover, previous investigations of the *in vitro* chemical degradation of polydioxanone [12,18] show that the Young's modulus of polydioxanone and radial force and pressure of the unloaded polydioxanone stent are maintained for up to 6–8 weeks. These findings prove that the force relaxation observed within the first 48 h is caused by viscoelasticity and not the chemical degradation of polydioxanone. To the best of our knowledge, the phenomenon of force relaxation due to polydioxanone's viscoelasticity has not previously been studied or published.

Despite our findings, we agree with other studies (*e.g.*, Cerna *et al.* [9], Repici *et al.* [10], Vandenplas *et al.* [11], and Zilberman *et al.* [12]) that using biodegradable stents is profitable and the right direction for the future of modern clinical practice. Nevertheless, stent force relaxation is a serious and clinically relevant problem. We suggest some possible solutions to overcome this problem.

The *in vitro* chemical degradation studies [12,18] suggest, and the results of *in vivo* studies [9–11] in which no premature stent collapse was observed show that (after the first 48 h) the overall radial force and hence the radial pressure of the implanted stent is sufficiently stable for at least 6 weeks. To predict the force relaxation caused by polydioxanone's viscoelasticity after 48 h when permanently loaded, we used the generalized Maxwell model, fit to our data. The relaxation time constants $\tau_1 = 4.50$ min and $\tau_2 = 189.9$ min confirm that most force relaxation occurs within the first two hours. The predicted axial force at infinite time, $F_\infty = 3.76$ N is close to the measured force after 48 h, 3.75 (3.36, 4.08) N, indicating that after 48 h, the subsequent force relaxation caused by polydioxanone's viscoelasticity is insignificant. Considering these findings, the force relaxation after 48 h is assumed to be clinically insignificant and compensated for by self-reinforcement due to polydioxanone's hydrophilicity; the force is stabilized. For this stabilized force, we propose the term "therapeutic force" and agree with the manufacturer's statement: "Stent integrity and radial force [therapeutic force] of the stent is maintained for 6–8 weeks following implantation. . .". Therefore, the design of a stent that sufficiently supports the treated tissue after stabilization is a possible solution. However, an initial force of approximately 50% higher than the therapeutic force of such a stent could be problematic, and the clinical impact of this issue should be investigated. Additionally, the development and use of more advanced biodegradable shape-memory materials should be considered.

A complete knowledge of biodegradable material properties and their limitations does not disqualify these materials but allows better design and clinical utility.

## Conclusions

We measured important properties of the SX-ELLA stent material, which we demonstrated to be a polydioxanone monofilament. The median Young's modulus of the filament was $\tilde{E} = 958$ (922, 974) MPa, and the shear modulus was $\tilde{G} = 357$ (185, 387) MPa, resulting in Poisson's ratio $\nu = 0.34$.

We confirmed our hypothesis that the SX-ELLA stents exhibit significant force relaxation due to polydioxanone monofilament stress relaxation, approximately 19% after the first 10 min and approximately 36% at 48 h after stent application.

However, these results were expected, and the manufacturer and implanting clinician should be aware of the known behavior of these biodegradable materials. If possible, a biodegradable stent should be designed considering therapeutic force rather than initial force. Additionally, new and more advanced shape-memory biodegradable polymers should be considered for future study and use.

## Supporting information

**S1 File.**
(PDF)

**S2 File.**
(ZIP)

**S3 File.**
(ZIP)

**S4 File.**
(ZIP)

**S5 File.**
(ZIP)

## Author Contributions

**Conceptualization:** Ales Bezrouk, Jan Kremlacek.

**Data curation:** Ales Bezrouk, Ludek Hromadko, Zuzana Olmrova Zmrhalova, Martin Kopecek, Martin Smutny, Jan M. Macak.

**Formal analysis:** Ales Bezrouk, Ludek Hromadko, Zuzana Olmrova Zmrhalova, Martin Kopecek, Martin Smutny, Iva Selke Krulichova, Jan M. Macak.

**Investigation:** Ales Bezrouk, Tomas Hosszu, Martin Kopecek.

**Methodology:** Ales Bezrouk, Tomas Hosszu, Martin Kopecek.

**Supervision:** Ales Bezrouk, Jan Kremlacek.

**Validation:** Ales Bezrouk, Tomas Hosszu, Jan Kremlacek.

**Writing – original draft:** Ales Bezrouk, Tomas Hosszu, Zuzana Olmrova Zmrhalova, Martin Kopecek, Martin Smutny, Iva Selke Krulichova, Jan M. Macak, Jan Kremlacek.

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
