## [Decision Letter · Decision Letter 0]

22 Apr 2020

PONE-D-20-05436

Mechanical properties of biodegradable self-expandable stent made of polydioxanone monofilament: *in vitro* force relaxation and its clinical relevance.

PLOS ONE

Dear Dr. Bezrouk,

Thank you for submitting your manuscript to PLOS ONE. After careful consideration, we feel that it has merit but does not fully meet PLOS ONE’s publication criteria as it currently stands. Therefore, we invite you to submit a revised version of the manuscript that addresses the points raised during the review process.

The reviewers agree on the significance of your proposed technique but have raised concerns regarding missing detail in the experiments and methods. Please address the reviewers' concerns and provide a point-by-point rebuttal. I recommend that you have your article proof-read one more time. Please also address the issue of data availability since it is not fully clear how you plan to make it available. 

We would appreciate receiving your revised manuscript by Jun 06 2020 11:59PM. To enhance the reproducibility of your results, we recommend that if applicable you deposit your laboratory protocols in protocols.io, where a protocol can be assigned its own identifier (DOI) such that it can be cited independently in the future. For instructions see: http://journals.plos.org/plosone/s/submission-guidelines#loc-laboratory-protocols

We look forward to receiving your revised manuscript.

Kind regards,

Jit Muthuswamy

Academic Editor

PLOS ONE

Additional Editor Comments:

The reviewers agree on the significance of your technique but have raised several issues regarding lack of detail in your experiments and methods. Please address these concerns in your revisions and provide a point-by-point rebuttal for these concerns. In addition, please also address how you plan to make your data available.

'The authors acknowledge the financial support from the Ministry of Education, Youth and Sports

of the Czech Republic (programme UK PROGRES Q40-09 and projects LQ1601 and

CZ.02.1.01/0.0/0.0/17_048/0007421).'

'The author(s) received no specific funding for this work.'

Reviewers' comments:

Reviewer's Responses to Questions

**Comments to the Author**

1. Is the manuscript technically sound, and do the data support the conclusions?

Reviewer #1: Partly

Reviewer #2: Partly

2. Has the statistical analysis been performed appropriately and rigorously? 

Reviewer #1: N/A

Reviewer #2: Yes

3. Have the authors made all data underlying the findings in their manuscript fully available?

Reviewer #1: No

Reviewer #2: Yes

4. Is the manuscript presented in an intelligible fashion and written in standard English?

Reviewer #1: No

Reviewer #2: Yes

5. Review Comments to the Author

Reviewer #1: The main purpose of this paper is to highlight important mechanical properties of a commercially-available, biodegradable stent (ELLA-SX) and discuss them for clinical relevance. However, the presented study is weak due to a lack of detail in the experimental protocol.

1. For SEM analysis were samples coated? Can you describe sample preparation?

2. It would be useful to show derivation for equation 1 from Zahora et al

3. In Equation 1 & 2 how does the Ln differ from Ls?

4. Extol 3426 (company, location)?

5. Briefly describe the measurement process for young’s modulus for the ELLA-SX stent based on kresinger and greenwald

6. Can the authors describe what the sample wires are and how they were acquired from the stent to determine stent elastic and shear modulus properties?

7. Captions for Figure 5 &6 could be more descriptive. In figure 5 what do the y axes represent? What is the inset represent in figure 6? How do the authors know this is a dioxanone monomer? Was this sample compared to dioxanone monomers?

8. Pg.18 “The obtained results from TGA analysis correspond to typical values for polydioxanone.” Can the authors offer some literature citations?

9. Can the authors quantify what is an appropriate therapeutic force? Can the authors clarify the statement on pg.22:

“An initial force higher than a therapeutic force by 50% (where the therapeutic force is considered to be 100%) means a therapeutic force lower than an initial force by 33.3% (where the initial force is considered to be 100%).”

10. The polymer of interest (PDX (polydioxannone)) is known to degrade over time in in vivo conditions. Can the authors comment on whether the measured force relaxation over 48 hours could be due to degradation of the polymer? How does elastic modulus change with degradation?

11. Did the authors model the viscoelastic properties to derive the creep resistance time constants ? How does the viscoelastic parameters change with time in vivo?

Minor comments

1. Overall, the grammar and punctuation throughout the manuscript needs to be addressed. There are several instances where commas are needed. For instance, in the abstract: “ Commercially available biodegradable self-expandable SX-ELLA stents..” needs commas. Another example on pg.24 “new and more advanced shape-memory biodegradable polymers may be considered..” requires commas.

2. Figure 1 and 2 captions have multiple dots as punctuation

3. Figure 3 caption needs more description.

4. Figure 7 and 8 captions have sentence fragments “Measured in an air-conditioned…”

5. Stand alone sentence (pg.22)

Reviewer #2: Reviewer Comments for PONE-D-20-05436 “Mechanical properties of a biodegradable self-expandable stent made of polydioxanone monofilament: in vitro force relaxation and its clinical relevance.”

The authors present a research study where they evaluate the mechanical properties of commercially biodegradable stents made of polydioxanone. They have developed a method the measure the shear modulus of small samples and evaluated the force relaxation behavior of stents.

I have some suggestions that the authors may want to address in order to improve the quality and readability of this manuscript.

1) TGA: why were some samples measured in air and others in a nitrogen atmosphere? Please comment. Also, there is no comment on the results of the TG-MS in the manuscript. Please add.

2) It might be helpful for the reader if the parameter LH would be added to figure 1.

3) Please provide more details for the determination of Young’s modulus instead of just pointing to references. Some actual stress-strain curves could be shown in the supplemental part.

4) Shear modulus measurements: it would be helpful if you could give more details about the tool that you have developed for the shear measurements. Since this tool is an essential part of this manuscript, it should be better visualized. Some photographs of the instrument and maybe a video in the supplemental part would help the reader to better understand how the tool looks like and how everything was set up with the Instron. Dimensions in Figure 2 are missing as well.

5) What were the dimensions of the wires used for mechanical measurements? How was it received? By disassembling the stents? Please provide some information in the manuscript.

6) In vitro force relaxation test: It is mentioned that the stents were extended by 56 mm. Was this extension longitudinal or radially? Please give further details. If the extension was longitudinal, what was the rationale for this? Why was 56 mm selected and what percentage of original dimensions does this represent? What was the radius before and after deformation? Please provide more details for this measurement.

7) To fully characterize polydioxanone, it would be good to see DSC and DMA measurements as well. The glass transition temperature, as well as storage and loss moduli, are important parameters to know. Displaying DMA curves also gives a good understanding of the viscoelastic behavior of the material.

8) The shear modulus was calculated from the experiments performed; thus, I am wondering why this is an absolute value and not error is given?

9) I would like to see some comments in the manuscript on the conditions used for the measurements. The force relaxation was measured in air with low humidity at body temperature, however, in vivo applications would not be dry. The materials properties might change upon contact with body fluids, they could plasticize. Young's modulus and shear were measured at room temperature I assume, but this also does not represent in vivo conditions.

10) You emphasize on the biodegradability of the stents. Thus, it would be worth investigating how the mechanical properties change with ongoing degradation and until what time after implantation the mechanical properties are still good enough to support the tissue. If this is beyond the scope f this manuscript, I would like to see at least a paragraph in the discussion or conclusion where the authors reflect on this topic.

6. PLOS authors have the option to publish the peer review history of their article (what does this mean?). If published, this will include your full peer review and any attached files.

Reviewer #1: No

Reviewer #2: No

---

## [Author Response · Author response to Decision Letter 0]

26 May 2020

First, we would like to thank professor Muthuswamy for the opportunity to revise our manuscript, titled "Mechanical properties of a biodegradable self-expandable polydioxanone monofilament stent: in vitro force relaxation and its clinical relevance.", with the help of the reviewers comments. The answers to prof. Muthuswamy questions and requirement, including the journal requirement are listed in the file "Cover Letter.docx" as requested.

We would like to thank the reviewers for their valuable reviews and comments, which, we believe, helped us to improve our manuscript significantly. Our amendments with regard to the reviewers' requests, including the links to relevant pages, are listed in the file "Response to Reviewers.docx" as requested by the journal (email form prof. Muthuswamy).

Plain text contained in the file "Cover Letter.docx":

Professor Jit Muthuswamy,

Hradec Králové, May 26, 2020

Dear professor Muthuswamy.

First, we would like to thank you for the opportunity to revise our manuscript, titled "Mechanical properties of a biodegradable self-expandable polydioxanone monofilament stent: in vitro force relaxation and its clinical relevance.", with the help of the reviewers comments. We also would like to thank the reviewers for their valuable reviews and comments, which, we believe, helped us to improve our manuscript significantly. Our amendments with regard to the reviewers' requests, including the links to relevant pages, are listed it the “Response to Reviewers.docx” file.

We followed your recommendation and resubmit our manuscript for a new proofreading. The original manuscript proofreading by American Journal Experts (AJE) has the verification code 2AF7-9CB9-E0B6-1B6A-00E7. The new proofreading (including the newly added text) by AJE has the verification code 974F-A6F6-3649-DC26-33AC. Our manuscript including the tracked changes is in the “Revised_Manuscript_with_Track_Changes.pdf”. The modified added and deleted text is highlighted in red color. The following language and grammer corrections are highlighted in blue.

In the following text on the next page, we response to the journal requirements.

We also followed your requirement to address the issue of data availability. We provide all the raw data, certain data processing samples and additional information (video and picture files) in the supporting information file S1 – S5, which are addressed in our manuscript.

We believe that all these requirements helped us to improve our manuscript significantly to make it worth publishing in your journal.

Thank you very much for your consideration. We look forward to your decision.

Yours sincerely,

Aleš Bezrouk, PhD

Department of Medical Biophysics, Faculty of Medicine in Hradec Kralove, Charles University

Czech Republic

Tel.: +420 495 816 255, E-mail: bezrouka@lfhk.cuni.cz

 

• Changed filenames of figures

• Changed figures naming

• Changed equation paragraph styles

• Modified author affiliations style

• Modified Corresponding author info

'The authors acknowledge the financial support from the Ministry of Education, Youth and Sports

of the Czech Republic (programme UK PROGRES Q40-09 and projects LQ1601 and

CZ.02.1.01/0.0/0.0/17_048/0007421).'

• Funding related text removed

a. Please clarify the sources of funding (financial or material support) for your study. List the grants or organizations that supported your study, including funding received from your institution.

• We acknowledge the financial support from the Ministry of Education, Youth and Sports of the Czech Republic:

o programme UK PROGRES Q40-09

o project LQ1601

o project CZ.02.1.01/0.0/0.0/17_048/0007421

o project LM2018103

• programme UK PROGRES Q40-09: Ales Bezrouk, Martin Kopecek, Iva Selke Krulichova

• project LQ1601: Jan Macak

• project CZ.02.1.01/0.0/0.0/17_048/0007421: Ales Bezrouk, Ludek Hromadko, Jan Macak

• project LM2018103: Jan Macak, Ludek Hromadko, Zuzana Olmrova Zmrhalova

• Figures formats checked and corrected

Plain text contained in the file "Response to Reviewers.docx":

 

We would like to thank the reviewers for their valuable reviews and comments, which, we believe, helped us to improve our manuscript significantly. Our amendments with regard to the reviewers' requests, including the links to relevant pages, are listed below.

 

Reviewer #1: The main purpose of this paper is to highlight important mechanical properties of a commercially-available, biodegradable stent (ELLA-SX) and discuss them for clinical relevance. However, the presented study is weak due to a lack of detail in the experimental protocol.

1. For SEM analysis were samples coated? Can you describe sample preparation?

 This is already described on page 6, however, we modified the text to make it clearer: The samples were prepared by mechanically rupturing the filaments by simply cracking them with tweezers, mounting them on the SEM stub using carbon adhesive tape, and directly (without an additional coating to prevent charging) visualizing them using field-emission SEM (FE-SEM JEOL JSM 7500F, Japan) and the in-lens secondary electron detector.

 Added; page 5: For the Young’s and shear moduli estimations, TGA, TGA-GCMS, and SEM, we acquired all the samples (sample wires) from new stents by removing the funnel-shaped ends and disassembling. 

2. It would be useful to show derivation for equation 1 from Zahora et al

 The complete derivation is already published by Zahora et al. It is not trivial and involves 4 pages of explanations, 15 equations and 3 schemes till Zahora et al. reaches the final derivation of pR and FR. We think that a complete derivation is beyond the scope of our publication, therefore we used only the two most important equations by Zahora et al. and modified them for our manuscript. Nevertheless, to make it clearer, we modified the text on page 6: “Then, using the approach and modifying the equations of Zahora et al. …” and provide some more detailed derivations in supporting information file (reference on page 7): A detailed derivation of Equations (1) and (2) is presented in S1 Derivation of Equations.

3. In Equation 1 & 2 how does the Ln differ from Ls?

 Clarified in the text on page 7: 

 “…and instantaneous total length LS.”

 Furthermore, in Equations (1) and (2), the total length of the stent LS is a measured parameter that changes with the applied load, while the nominal length of the stent LN and the helix length LH are the nominal parameters of an unloaded stent, which remain constant during measurement. 

4. Extol 3426 (company, location)?

 Added; page 7, 8, 9, 10: (Madal Bal a. s., Czech Republic)

5. Briefly describe the measurement process for young’s modulus for the ELLA-SX stent based on kresinger and greenwald

 Added text; page 7, 8: We gathered the force-extension data, which we then further transformed to standard stress-strain curves using the sample wire diameter d and the original effective length of the sample LTS. We performed 20 measurements of LTS using the Extol 3426 caliper (Madal Bal a. s., Czech Republic). We determined the Young’s modulus as the slope of the greatest pseudolinear part of the stress-strain curve.

6. Can the authors describe what the sample wires are and how they were acquired from the stent to determine stent elastic and shear modulus properties?

 Added; page 5: For the Young’s and shear moduli estimations, TGA, TGA-GCMS, and SEM, we acquired all the samples (sample wires) from new stents by removing the funnel-shaped ends and disassembling. 

7. Captions for Figure 5 &6 could be more descriptive. In figure 5 what do the y axes represent? What is the inset represent in figure 6? How do the authors know this is a dioxanone monomer? Was this sample compared to dioxanone monomers?

 The Figure 5 was modified to be more intelligible and self-explanable (including the Y axes)

 Figure 5 caption modified: Fig 5. TGA results for the 3rd monofilament stent wire sample. The black line shows the dependence of the relative sample weight on the temperature. The blue dashed line shows the dependence of the rate of the change of the sample weight (derivate weight) on the temperature. The derivative thermal gravimetric (DTG) peak is observed at 234°C.

 The Figure 6 was modified (units of the X axis)

 Text modified, page 15: During TGA of the 3rd sample, the released gas was taken to a gas chromatograph coupled to a mass spectrometer using a heated transfer line. The chromatogram of the 3rd sample (Fig 6) shows a single peak, which means that only a single substance was detected during the thermal degradation of the sample. Subsequently, the molecular ion peak in the measured mass spectrum shown in the inset of Fig 6 has a value of m/z = 102, which corresponds to the molecular weight of a dioxanone monomer.

 Figure 6 caption modified: Fig 6. Chromatogram and mass spectrum (shown in the inset) for the 3rd monofilament stent wire sample from GCMS analysis. The red curve in the chromatogram shows the relative intensity, with a single peak detected at 4.33 min. The maximum relative intensity (100%) is derived from the maximum peak, which corresponds to the carrier gas N2 (not shown). The red vertical lines in the mass spectrum in the inset show the relative intensity for the detected ions. The maximum relative intensity (100%) is derived from the maximum peak m/z = 28, which corresponds to the carrier gas N2. The molecular ion peak has a value of m/z = 102, which corresponds to the molecular weight of a dioxanone monomer.

8. Pg.18 “The obtained results from TGA analysis correspond to typical values for polydioxanone.” Can the authors offer some literature citations?

 Added references, page 14: The TGA results correspond to previously published values for polydioxanone by Wang et al. [32] and Huang et al. [33].

 Added text, page 18: The TGA results for the 3rd sample measured in a nitrogen atmosphere corresponds to the data published by Wang et al. [32] and Huang et al. [33], who used a similar experimental setup and a heating rate of 10 °C/min. However, other studies [34,35] show different TGA results, which may be caused by a different experimental setup, heating rate or material type (e.g., lower molecular weight polymer chains) and its possible postproduction processing.

9. Can the authors quantify what is an appropriate therapeutic force? 

 We are sorry but the quantifying of an “appropriate” therapeutic force is a different question and it is a very complex task and definitely beyond the scope of this study. It depends on many factors such as patient’s anatomy, specific diagnosis, specific treatment and it is the sole responsibility of the physician and his choice of which force is appropriate for his patient. Then, according the desired therapeutic force and other parameters, the physician will choose an appropriate stent. The radial forces (not the therapeutic forces) of the specific stent are provided (should be provided) by the producers of the stents.

 Nevertheless, the absolutely key information provided by this study is, that this therapeutic force, however great or small, must be constant during the desired time of treatment. This study found and points out a great decrease of the therapeutic force of biodegradable stents during the first 48 hours after application, which is never published information, and suggests some possible solutions.

 Therefore, the physician should be aware of this issue and for instance, if possible, choose a stent with 50% higher initial force than usually. 

 To be clearer and more intuitive, we added text on page 18 (Discussion): Stenting is an important medical method in which the implanted stent exerts a force on the surrounding tissue to reopen occluded hollow tubular body structures. In clinical practice, considering many factors, such as the patient’s anatomy, specific diagnosis, and specific treatment, it is the sole responsibility of the physician to select a force appropriate for the patient. Then, with respect to the desired force, the physician selects a suitable stent with the appropriate parameters. Therefore, the parameters of the stent and its material are crucial.

Can the authors clarify the statement on pg.22: “An initial force higher than a therapeutic force by 50% (where the therapeutic force is considered to be 100%) means a therapeutic force lower than an initial force by 33.3% (where the initial force is considered to be 100%).”

 This explanation was mentioned as a more detailed description or clarification of the previous sentence: “However, a higher initial force of approximately 50% compared with the therapeutic force …”. Since this sentence is clear to both the reviewers and the above explanation was originally suggested by only one our co-author, we deleted this explanation because, obviously, it has now lost its meaning.

10. The polymer of interest (PDX (polydioxannone)) is known to degrade over time in in vivo conditions. Can the authors comment on whether the measured force relaxation over 48 hours could be due to degradation of the polymer? How does elastic modulus change with degradation?

 Text added, page 19, 20: To avoid chemical degradation of the stent material, and considering the storage conditions recommended by the stent manufacturer, we performed all the mechanical measurements in an environment with a low relative humidity (approx. 30%). Furthermore, to eliminate the effect of temperature, we performed all the mechanical measurements at a constant “body” temperature of (37 ± 0.3)°C. The TGA in an air atmosphere showed that the polydioxanone filament is stable at 37°C and does not thermally degrade. This allowed us to separately study the effect of stent material viscoelasticity on the radial force and pressure. Moreover, previous investigations of the in vitro chemical degradation of polydioxanone [12,18] show that the Young’s modulus of polydioxanone and radial force and pressure of the unloaded polydioxanone stent are maintained for up to 6 – 8 weeks. These findings prove that the force relaxation observed within the first 48 h is caused by viscoelasticity and not the chemical degradation of polydioxanone. To the best of our knowledge, the phenomenon of force relaxation due to polydioxanone’s viscoelasticity has not previously been studied or published. 

11. Did the authors model the viscoelastic properties to derive the creep resistance time constants ? 

 Added text, page 11, 12 (M&M): For verification and prediction, we also modeled the force relaxation using the generalized Maxwell model, described by Equation (7):

 F_A (t)=F_∞+∑_n▒〖F_n e^(-t/τ_n ) 〗 (7)

where FA represents the relaxation of the stent’s axial force over time, t is the relaxation time, F∞ is the axial force of the stent at infinite time, i.e., the theoretical steady axial force after the end of the relaxation process, n is the number of partial processes contributing to the total axial force relaxation, Fn is the magnitude of the contribution of the partial process to the total relaxation, and τn is the relaxation time constant of the partially contributing process. 

 Added text, page 17 (Results): We obtained the best fit to the relaxation data using the generalized Maxwell model (Fig 8) assuming two partially contributing processes. The primary relaxation process was fast, with a relaxation time constant τ1 = 4.50 min and a relaxation magnitude F1 = 1.19 N. The secondary relaxation process was slower, with a relaxation time constant τ2 = 189.9 min and a relaxation magnitude F2 = 0.90 N. The axial force at an infinite time limit was F∞ = 3.76 N.

 Modified Fig 8

 Modified Fig 8 caption: “…previous time step. The red plus sign (+) indicates the calculated force value using the generalized Maxwell model.”

How does the viscoelastic parameters change with time in vivo?

 Added text, page 20: The in vitro chemical degradation studies [12,18] suggest, and the results of in vivo studies [9-11] in which no premature stent collapse was observed show that (after the first 48 h) the overall radial force and hence the radial pressure of the implanted stent is sufficiently stable for at least 6 weeks. To predict the force relaxation caused by polydioxanone’s viscoelasticity after 48 h when permanently loaded, we used the generalized Maxwell model, fit to our data. The relaxation time constants τ1 = 4.50 min and τ2 = 189.9 min confirm that most force relaxation occurs within the first two hours. The predicted axial force at infinite time, F∞ = 3.76 N is close to the measured force after 48 h, 3.75 (3.36, 4.08) N, indicating that after 48 h, the subsequent force relaxation caused by polydioxanone’s viscoelasticity is insignificant. Considering these findings, the force relaxation after 48 h is assumed to be clinically insignificant and compensated for by self-reinforcement due to polydioxanone’s hydrophilicity; the force is stabilized.

Minor comments

1. Overall, the grammar and punctuation throughout the manuscript needs to be addressed. There are several instances where commas are needed. For instance, in the abstract: “ Commercially available biodegradable self-expandable SX-ELLA stents..” needs commas. Another example on pg.24 “new and more advanced shape-memory biodegradable polymers may be considered..” requires commas.

 The manuscript was already proofread by AJE 

 (verification code 2AF7-9CB9-E0B6-1B6A-00E7), 

 nevertheless, we resubmitted the manuscript for a new proofreading and left the language, punctuation, and overall grammar correction, including the new added text, to the decision of AJE (American Journal Experts) professionals

 (verification code 974F-A6F6-3649-DC26-33AC).

2. Figure 1 and 2 captions have multiple dots as punctuation

 Figure 1 caption modified: LN is the total nominal length of the unloaded stent; D1, D2 are the nominal stent diameters; LH is the length of a single turn of a single helix of the unloaded stent (helix length).

 Figure 2 caption modified: Fig 2. Photograph and scheme of the dedicated tool for shear modulus measurement. The device consists of a solid frame denoted by A, a fixed Jacobs microchuck denoted by B, a revolving Jacobs microchuck, denoted by C, tightly connected to a pair of pulleys, and a pulling wire, denoted by E, connected to the “pulling” pulley on one side and the Instron force transducer on the other side (marked by the arrow). The tested specimen (wire) is denoted by D. The thick gray dashed line denotes the elastic reverting element connected to the “reverting” pulley on one side and to the solid frame on the other side. DRP is the diameter of the “pulling” pulley; d is the stent wire diameter; LW is the specimen (wire) length.

3. Figure 3 caption needs more description. 

 Figure 3 caption changed: Fig 3. Details of the custom holding tool. The tool consists of a hollow cylinder with one open end and one closed end. A hook is mounted to the closed end, allowing its attachment to the measuring device. There are three small holes in the cylinder shell for the transverse attachment of stainless-steel needles.

 Added text, page 11: Each stent was fixed at its ends by a pair of custom holding tools, as shown in Fig 3. Each stent end was inserted into the tool’s cylinder cavity and fixed using the stainless-steel needles, which were threaded transversely through the meshes at the stent ends. This allowed the stent to move relatively freely and easily change its diameter without significant friction.

4. Figure 7 and 8 captions have sentence fragments “Measured in an air-conditioned…”

 Figure 7 and 8 captions modified: “The samples were measured …”

5. Stand alone sentence (pg.22)

We have not found any stand alone sentence on the page 22. However, we think that the reviewer #1 had this sentence in mind: Additionally, the development and use of more advanced biodegradable shape-memory materials should be considered. Then, to make it more intelligible, we moved this sentence at the end of the previous paragraph on page 21.

Reviewer #2: Reviewer Comments for PONE-D-20-05436 “Mechanical properties of a biodegradable self-expandable stent made of polydioxanone monofilament: in vitro force relaxation and its clinical relevance.” The authors present a research study where they evaluate the mechanical properties of commercially biodegradable stents made of polydioxanone. They have developed a method the measure the shear modulus of small samples and evaluated the force relaxation behavior of stents. I have some suggestions that the authors may want to address in order to improve the quality and readability of this manuscript.

1) TGA: why were some samples measured in air and others in a nitrogen atmosphere? 

 Added text in the M&M, page 5,6: During TGA, we heated the samples to 600 °C at a rate of 10 °C/min. The samples were measured in an open platinum crucible in a synthetic air atmosphere (20 ml/min). The samples were measured in an air atmosphere to determine polydioxanone’s thermal stability in this atmosphere. TGA was performed on a Pyris 1 TGA (PerkinElmer, USA). 

During TGA-GCMS, we heated the sample to 300 °C at a rate of 10 °C/min. The samples were measured in an open platinum crucible in a nitrogen atmosphere (20 ml/min; Linde N2 4.6). In TGA-GCMS, a nitrogen atmosphere must be used to protect the mass spectrometer’s heated wolfram filament from oxygen. The TGA-GCMS was performed on a Pyris 1 TGA, GC Clarus 680, and MS Clarus SQ 8T (all PerkinElmer, USA).

 Added text (Results), page 14: The TGA results correspond to previously published values for polydioxanone by Wang et al. [32] and Huang et al. [33]. We did not observe any decrease in thermal stability of the polydioxanone filament in a synthetic air atmosphere compared to a nitrogen atmosphere.

 Added text (Discussion), page 19: To avoid chemical degradation of the stent material, and considering the storage conditions recommended by the stent manufacturer, we performed all the mechanical measurements in an environment with a low relative humidity (approx. 30%). Furthermore, to eliminate the effect of temperature, we performed all the mechanical measurements at a constant “body” temperature of (37 ± 0.3)°C. The TGA in an air atmosphere showed that the polydioxanone filament is stable at 37°C and does not thermally degrade. This allowed us to separately study the effect of stent material viscoelasticity on the radial force and pressure.

Please comment. Also, there is no comment on the results of the TG-MS in the manuscript. Please add.

 Added text, page 15: During TGA of the 3rd sample, the released gas was taken to a gas chromatograph coupled to a mass spectrometer using a heated transfer line. The chromatogram of the 3rd sample (Fig 6) shows a single peak, which means that only a single substance was detected during the thermal degradation of the sample. Subsequently, the molecular ion peak in the measured mass spectrum shown in the inset of Fig 6 has a value of m/z = 102, which corresponds to the molecular weight of a dioxanone monomer.

 Figure 6 modified

 Figure 6 caption modified: Fig 6. Chromatogram and mass spectrum (shown in the inset) for the 3rd monofilament stent wire sample from GCMS analysis. The red curve in the chromatogram shows the relative intensity, with a single peak detected at 4.33 min. The maximum relative intensity (100%) is derived from the maximum peak, which corresponds to the carrier gas N2 (not shown). The red vertical lines in the mass spectrum in the inset show the relative intensity for the detected ions. The maximum relative intensity (100%) is derived from the maximum peak m/z = 28, which corresponds to the carrier gas N2. The molecular ion peak has a value of m/z = 102, which corresponds to the molecular weight of a dioxanone monomer.

 Added text, page 18: All the findings from SEM, TGA, and TGA-GCMS agree with the manufacturer’s information on the product label, i.e., the stent is made of a single polydioxanone monofilament. The TGA results for the 3rd sample measured in a nitrogen atmosphere corresponds to the data published by Wang et al. [32] and Huang et al. [33], who used a similar experimental setup and a heating rate of 10 °C/min. However, other studies [34,35] show different TGA results, which may be caused by a different experimental setup, heating rate or material type (e.g., lower molecular weight polymer chains) and its possible postproduction processing.

2) It might be helpful for the reader if the parameter LH would be added to figure 1.

 Figure 1 modified (LH parameter added)

 Figure 1 caption modified accordingly

3) Please provide more details for the determination of Young’s modulus instead of just pointing to references. Some actual stress-strain curves could be shown in the supplemental part.

 An example of processed data of the sample1 and a figure of the stress strain curve with the calculated Young’s modulus are provided in supporting information file; page 16: We provide the raw data, an example of processed data for Sample 1, and the stress-strain curve for Sample 1 in S4 Tensile Data.

 All the raw data are provided in supporting information files; page 16: We provide the raw data, an example of processed data for Sample 1, and the stress-strain curve for Sample 1 in S4 Tensile Data.

 Added text; page 7, 8: We gathered the force-extension data, which we then further transformed to standard stress-strain curves using the sample wire diameter d and the original effective length of the sample LTS. We performed 20 measurements of LTS using the Extol 3426 caliper (Madal Bal a. s., Czech Republic). We determined the Young’s modulus as the slope of the greatest pseudolinear part of the stress-strain curve.

4) Shear modulus measurements: it would be helpful if you could give more details about the tool that you have developed for the shear measurements. Since this tool is an essential part of this manuscript, it should be better visualized. Some photographs of the instrument and maybe a video in the supplemental part would help the reader to better understand how the tool looks like and how everything was set up with the Instron. Dimensions in Figure 2 are missing as well.

 Added tool photograph into the Figure 2

 Added missing dimensions and modified caption of Figure 2: Fig 2. Photograph and scheme of the dedicated tool for shear modulus measurement. The device consists of a solid frame denoted by A, a fixed Jacobs microchuck denoted by B, a revolving Jacobs microchuck, denoted by C, tightly connected to a pair of pulleys, and a pulling wire, denoted by E, connected to the “pulling” pulley on one side and the Instron force transducer on the other side (marked by the arrow). The tested specimen (wire) is denoted by D. The thick gray dashed line denotes the elastic reverting element connected to the “reverting” pulley on one side and to the solid frame on the other side. DRP is the diameter of the “pulling” pulley; d is the stent wire diameter; LW is the specimen (wire) length.

 Added supporting information files with additional photos and videos (reference on page 9): An overview with detailed photos and videos of the dedicated tool with the measured specimen are provided in S2 Tool Photos and Videos.

5) What were the dimensions of the wires used for mechanical measurements? How was it received? By disassembling the stents? Please provide some information in the manuscript.

 Added; page 5: For the Young’s and shear moduli estimations, TGA, TGA-GCMS, and SEM, we acquired all the samples (sample wires) from new stents by removing the funnel-shaped ends and disassembling. 

 The wire diameter is now obviously the stent wire diameter d, which is already described in the paragraph “Stent geometry”

 The specimen length LW is already described in the paragraph “Shear modulus” 

6) In vitro force relaxation test: It is mentioned that the stents were extended by 56 mm. Was this extension longitudinal or radially? Please give further details. 

 Added text (M&M), page 11: Then, each stent was extended by 56 mm (+70%) from its nominal length (LN = 80 mm) to a total length of LS = 136 mm, at 10 mm·s-1 and held in the extended position for at least 48 h (172 800 s) at a constant temperature (37 ± 0.3)°C.

If the extension was longitudinal, what was the rationale for this? 

 Added text (M&M), page 6: Knowledge of stent geometry parameters is important for design, clinical practice, and further mechanical analyses. From a clinical perspective, the most important information is the dependence of stent radial pressure pR on stent deformation. The radial force FR is then used as an important parameter for stent production output quality control. However, it is difficult to accurately measure these radial parameters. Typical significant problems with direct radial measurements include the mechanical interaction between the measuring device and the stent and the effect of friction. These negative effects cannot be sufficiently determined and thus effectively and accurately compensated for. Instead, the easier and more accurate measurements of stent axial force and deformation can be used [28]. The effects of friction and device-stent interactions in the axial measurements are negligible. The measured data can then be mathematically transformed into clinically relevant radial pressure and deformation data.

Why was 56 mm selected and what percentage of original dimensions does this represent? What was the radius before and after deformation? Please provide more details for this measurement.

 Text added (M&M, page 10,11): “The purpose of the in vitro force relaxation tests was to separately study the effect of the viscoelasticity of the stent material on the radial force and pressure with respect to esophageal anatomy. The normal esophagus diameter in the human population is approximately 20 – 33 mm [31]. A symptomatic benign stenosis is clinically variable but is often under 13 mm and rarely over 20 mm in diameter [31]. Therefore, we chose 17 mm as an average symptomatic diameter of an esophageal stricture. An axial stent elongation of 56 mm caused a change in our stent diameter from the nominal 25 mm to the desired 17 mm. The relationship between the length and the diameter of a braided stent was published by Zahora et al. [28], and it is also shown in S1 Derivation of Equations. The force-relaxation tests were performed on an Instron 3343 …”

7) To fully characterize polydioxanone, it would be good to see DSC and DMA measurements as well. The glass transition temperature, as well as storage and loss moduli, are important parameters to know. Displaying DMA curves also gives a good understanding of the viscoelastic behavior of the material.

 The main purpose of our investigation was to study how much the viscoelasticity of the stent material (excluding the other effects like chemical degradation) affects its ability to support the threated tissue and discuss its clinical relevance (e.g., possible premature stent collapse). To our best knowledge, the effect of force relaxation of a permanently loaded polydioxanone stent was never studied and published. The SEM, TGA, and TGA-GCMS analyses were used as additional analyses in order to verify whether the used material of the stent is a simple (not composite) polydioxanone monofilament. The TGA, DMA, DSC, storage modulus, complex viscosity, FT-IR spectrums, WAXD patterns, chemical degradation of an unloaded polydioxanone (stent) (Young’s modulus, radial strength, pH measurements) analyses results were already published. Therefore, we added the relevant references in the text and discussed them:

 Added references, page 14: The TGA results correspond to previously published values for polydioxanone by Wang et al. [32] and Huang et al. [33].

 Added text, page 19: All the findings from the SEM, TGA, and TGA-GSMC analyses agree with the manufacturer’s information on the product label, i.e., the stent is made of a single polydioxanone monofilament. All the findings from SEM, TGA, and TGA-GCMS agree with the manufacturer’s information on the product label, i.e., the stent is made of a single polydioxanone monofilament. The TGA results for the 3rd sample measured in a nitrogen atmosphere corresponds to the data published by Wang et al. [32] and Huang et al. [33], who used a similar experimental setup and a heating rate of 10 °C/min. However, other studies [34,35] show different TGA results, which may be caused by a different experimental setup, heating rate or material type (e.g., lower molecular weight polymer chains) and its possible postproduction processing.

 Added text, page 19: For technical reference, certain additional rheological properties of polydioxanone can be found in the study published by Ahlinder et al. [34]. 

 Moreover, we are very sorry, but due to Covid19 situation causing limiting the producer’s capacity, we verified at the producer that it is impossible to obtain extra samples for measurement and test purposes at this moment. Unfortunately, but expectably, all the samples we had completely degraded and thus cannot be used. 

8) The shear modulus was calculated from the experiments performed; thus, I am wondering why this is an absolute value and not error is given?

 Added; page 2, 16, 19, 22: G ~ = 357 (185, 387) MPa 

9) I would like to see some comments in the manuscript on the conditions used for the measurements. The force relaxation was measured in air with low humidity at body temperature, however, in vivo applications would not be dry. The materials properties might change upon contact with body fluids, they could plasticize. 

 Text added, page 19,20: To avoid chemical degradation of the stent material, and considering the storage conditions recommended by the stent manufacturer, we performed all the mechanical measurements in an environment with a low relative humidity (approx. 30%). Furthermore, to eliminate the effect of temperature, we performed all the mechanical measurements at a constant “body” temperature of (37 ± 0.3)°C. The TGA in an air atmosphere showed that the polydioxanone filament is stable at 37°C and does not thermally degrade. This allowed us to separately study the effect of stent material viscoelasticity on the radial force and pressure. Moreover, previous investigations of the in vitro chemical degradation of polydioxanone [12,18] show that the Young’s modulus of polydioxanone and radial force and pressure of the unloaded polydioxanone stent are maintained for up to 6 – 8 weeks. These findings prove that the force relaxation observed within the first 48 h is caused by viscoelasticity and not the chemical degradation of polydioxanone. To the best of our knowledge, the phenomenon of force relaxation due to polydioxanone’s viscoelasticity has not previously been studied or published. 

 Added text, page 20: The in vitro chemical degradation studies [12,18] suggest, and the results of in vivo studies [9-11] in which no premature stent collapse was observed show that (after the first 48 h) the overall radial force and hence the radial pressure of the implanted stent is sufficiently stable for at least 6 weeks. To predict the force relaxation caused by polydioxanone’s viscoelasticity after 48 h when permanently loaded, we used the generalized Maxwell model, fit to our data. The relaxation time constants τ1 = 4.50 min and τ2 = 189.9 min confirm that most force relaxation occurs within the first two hours. The predicted axial force at infinite time, F∞ = 3.76 N is close to the measured force after 48 h, 3.75 (3.36, 4.08) N, indicating that after 48 h, the subsequent force relaxation caused by polydioxanone’s viscoelasticity is insignificant. Considering these findings, the force relaxation after 48 h is assumed to be clinically insignificant and compensated for by self-reinforcement due to polydioxanone’s hydrophilicity; the force is stabilized.

Young's modulus and shear were measured at room temperature I assume, but this also does not represent in vivo conditions.

 Added missing test conditions; page 7 and 9: We performed the measurements in a temperature-controlled chamber at a constant temperature of (37 ± 0.3)°C. We also monitored the relative humidity for extreme fluctuations, i.e., a difference of more than 10%, due to polydioxanone’s natural hydrophilicity that could potentially affect the measured data.

 Test conditions during the in-vitro test are already given; page 11.

10) You emphasize on the biodegradability of the stents. Thus, it would be worth investigating how the mechanical properties change with ongoing degradation and until what time after implantation the mechanical properties are still good enough to support the tissue. If this is beyond the scope f this manuscript, I would like to see at least a paragraph in the discussion or conclusion where the authors reflect on this topic.

 We are sorry but the in vivo study is beyond the scope of our study. Therefore, we extended the discussion with references addressing this issue: 

 Text added, page 19,20: To avoid chemical degradation of the stent material, and considering the storage conditions recommended by the stent manufacturer, we performed all the mechanical measurements in an environment with a low relative humidity (approx. 30%). Furthermore, to eliminate the effect of temperature, we performed all the mechanical measurements at a constant “body” temperature of (37 ± 0.3)°C. The TGA in an air atmosphere showed that the polydioxanone filament is stable at 37°C and does not thermally degrade. This allowed us to separately study the effect of stent material viscoelasticity on the radial force and pressure. Moreover, previous investigations of the in vitro chemical degradation of polydioxanone [12,18] show that the Young’s modulus of polydioxanone and radial force and pressure of the unloaded polydioxanone stent are maintained for up to 6 – 8 weeks. These findings prove that the force relaxation observed within the first 48 h is caused by viscoelasticity and not the chemical degradation of polydioxanone. To the best of our knowledge, the phenomenon of force relaxation due to polydioxanone’s viscoelasticity has not previously been studied or published. 

 Added text, page 20: The in vitro chemical degradation studies [12,18] suggest, and the results of in vivo studies [9-11] in which no premature stent collapse was observed show that (after the first 48 h) the overall radial force and hence the radial pressure of the implanted stent is sufficiently stable for at least 6 weeks. To predict the force relaxation caused by polydioxanone’s viscoelasticity after 48 h when permanently loaded, we used the generalized Maxwell model, fit to our data. The relaxation time constants τ1 = 4.50 min and τ2 = 189.9 min confirm that most force relaxation occurs within the first two hours. The predicted axial force at infinite time, F∞ = 3.76 N is close to the measured force after 48 h, 3.75 (3.36, 4.08) N, indicating that after 48 h, the subsequent force relaxation caused by polydioxanone’s viscoelasticity is insignificant. Considering these findings, the force relaxation after 48 h is assumed to be clinically insignificant and compensated for by self-reinforcement due to polydioxanone’s hydrophilicity; the force is stabilized.

---

## [Decision Letter · Decision Letter 1]

24 Jun 2020

Mechanical properties of a biodegradable self-expandable polydioxanone monofilament stent:*in vitro* force relaxation and its clinical relevance.

PONE-D-20-05436R1

Dear Dr. Bezrouk,

We’re pleased to inform you that your manuscript has been judged scientifically suitable for publication and will be formally accepted for publication once it meets all outstanding technical requirements.

Kind regards,

Jit Muthuswamy

Academic Editor

PLOS ONE

Additional Editor Comments (optional):

Reviewers' comments:

Reviewer's Responses to Questions

**Comments to the Author**

1. If the authors have adequately addressed your comments raised in a previous round of review and you feel that this manuscript is now acceptable for publication, you may indicate that here to bypass the “Comments to the Author” section, enter your conflict of interest statement in the “Confidential to Editor” section, and submit your "Accept" recommendation.

Reviewer #1: All comments have been addressed

Reviewer #2: All comments have been addressed

2. Is the manuscript technically sound, and do the data support the conclusions?

Reviewer #1: Yes

Reviewer #2: Yes

3. Has the statistical analysis been performed appropriately and rigorously? 

Reviewer #1: Yes

Reviewer #2: Yes

4. Have the authors made all data underlying the findings in their manuscript fully available?

Reviewer #1: Yes

Reviewer #2: Yes

5. Is the manuscript presented in an intelligible fashion and written in standard English?

Reviewer #1: Yes

Reviewer #2: Yes

6. Review Comments to the Author

Reviewer #1: (No Response)

Reviewer #2: Thanks for revising the manuscript according to the reviewers' comments. I am satisfied with the revision as all my concerns have been addressed.

7. PLOS authors have the option to publish the peer review history of their article (what does this mean?). If published, this will include your full peer review and any attached files.

Reviewer #1: No

Reviewer #2: No

---

## [Editor Report · Acceptance letter]

26 Jun 2020

PONE-D-20-05436R1 

Mechanical properties of a biodegradable self-expandable polydioxanone monofilament stent: *in vitro* force relaxation and its clinical relevance. 

Dear Dr. Bezrouk:

I'm pleased to inform you that your manuscript has been deemed suitable for publication in PLOS ONE. Congratulations! Your manuscript is now with our production department. 

Kind regards, 

on behalf of

Dr. Jit Muthuswamy 

Academic Editor

PLOS ONE